



# Gridded pollen-based Holocene regional plant cover in temperate and northern subtropical China suitable for climate modeling

Furong Li[1,2], Marie-José Gaillard[2], Xianyong Cao[3], Ulrike Herzschuh[4,5,6], Shinya Sugita [7], Jian Ni[8], Yan Zhao[9,10], Chengbang An[11], Xiaozhong Huang[11], Yu Li[11], Hongyan Liu[12], Aizhi Sun[10], Yifeng Yao[13]

[1]School of Ecology, Sun Yat-sen University, Shenzhen, 518107, China

[2]Department of Biology and Environmental Science, Linnaeus University, Kalmar 39182, Sweden

[3]Alpine Paleoecology and Human Adaptation Group (ALPHA), State Key Laboratory of Tibetan Plateau Earth System, and Resources and Environment (TPESRE), Institute of Tibetan Plateau Research, Chinese Academy of Sciences, Beijing 100101, China

[4]Alfred Wegener Institute Helmholtz Center for Polar and Marine Research, Research Unit Potsdam, Potsdam 14473, Germany

[5]Institute of Environmental Science and Geography, University of Potsdam, Potsdam 14476, Germany

[6]Institute of Biochemistry and Biology, University of Potsdam, Potsdam 14476, Germany

[7]Institute of Ecology, University of Tallinn, Tallinn 10120, Estonia

[8]College of Chemistry and Life Sciences, Zhejiang Normal University, Jinhua 321004, China

[9]Institute of Geographic Sciences and Natural Resources Research, Chinese Academy of Sciences, Beijing 100101, China

[10]University of Chinese Academy of Sciences, Beijing 100101, China

[11]College of Earth and Environmental Sciences, Lanzhou University, Lanzhou 730000, China

[12]College of Urban and Environmental Sciences and MOE Laboratory for Earth Surface Processes, Peking University, Beijing 100871, China

[13]Institute of Botany, Chinese Academy of Sciences, Beijing 100093, China

*Correspondence to*: Furong Li (lifr5@mail.sysu.edu.cn)



**Abstract.** We present the first gridded and temporally continuous quantitative pollen-based plant-cover reconstruction for temperate and northern sub-tropical China over the Holocene (11.7 ka BP to present) applying the Regional Estimates of Vegetation Abundance from Large Sites (REVEALS) model. The objective is to provide a dataset of pollen-based land cover for the last ca. twelve millennia suitable for palaeoclimate modeling and evaluation of simulated past vegetation cover from dynamic vegetation models and anthropogenic land-cover change (ALCC) scenarios. The REVEALS reconstruction was achieved using 94 selected pollen records from lakes and bogs at a 1˚×1˚ spatial scale and a temporal resolution of 500 years between 11.7 and 0.7 ka BP, and three recent time windows (0.7−0.35 ka BP, 0.35−0.1 ka BP, and 0.1 ka BP−present). The dataset includes REVEALS estimates of cover and their standard errors (SEs) for 27 plant taxa in 75 1˚×1˚ grid cells distributed within the study region. The 27 plant taxa were also grouped into six plant functional types and three land-cover types (coniferous trees CT, broadleaved trees BT, and C3 herbs C3H/open land OL), and their REVEALS estimates of cover and related SEs were calculated. We describe the protocol used for the selection of pollen records and the REVEALS application (with parameter setting), and explain the major rationales behind the protocol. As an illustration we present, for eight selected time windows, gridded maps of the pollen-based REVEALS estimates of cover for the three land-cover types (CT, BT, and C3H/OL). We then discuss the reliability and limitations of the Chinese dataset of Holocene gridded REVEALS plant-cover, and its current and potential uses.

The dataset is available at the National Tibetan Plateau Data Center (TPDC; https://data.tpdc.ac.cn/en/disallow/d18d2b7e-25fe-49da-b1bd-2be6014162b0/.)



## Introduction


Vegetation has undergone changes over the globe during the entire Holocene as a result of climate change from
the early Holocene and disturbance from anthropogenic activities from the mid Holocene (e.g. ArchaeoGLOBE,
2019;  Li et al., 2020; Marquer et al., 2017). Pollen- data mapping can provide insights on temporal and spatial
vegetation change at broad continental scales (Huntley and Birks, 1983; Huntley and Iii., 1988; Ren and Zhang,
1998; Ren and Beug, 2002). However, quantification of past vegetation change based on fossil pollen data is
necessary for specific research questions on the relationship between plant cover and e.g. climate or biodiversity.
Techniques such as biomization (Prentice and Webb Iii, 1998) and Modern Analog Technique (MAT) (Overpeck
et al., 1985) were widely applied to reconstruct past continental-scale changes in vegetation cover. These
techniques have the disadvantage that they cannot quantify the cover of individual plant taxa. In this paper, we
present the first pollen-based quantitative reconstruction of Holocene plant-cover change in temperate and northern
subtropical China using the Regional Estimates of VEgetation Abundance from Large Sites (REVEALS) model
(Sugita, 2007a).
The possible effects of anthropogenic land-cover (LC) transformation due to past land-use (LU) change (LULCs)
on Holocene climate is still an issue of debate (Harrison et al., 2020). Current earth system models (ESMs) take
care of the climate–land vegetation interactions by coupling a dynamic vegetation model (DVM) with the climate
model (e.g. Claussen et al., 2013; Lu et al., 2018; Wyser et al., 2020). DVMs simulate climate-induced (natural)
vegetation. Therefore, estimates of past LULCs have to be estimated to study their effect on past climate. The
anthropogenic land-cover change scenarios (ALCCs) most commonly used by palaeoclimate modelers are those
from the HYDE database (Klein Goldewijk et al., 2017) and the KK10 dataset of past deforestation (Kaplan et al.,
2009). These scenarios are based on a number of assumptions on population growth, per-capita land use, and other
parameters influencing land use over time in the past (e.g. Kaplan et al. 2017). Therefore, a current priority is to
produce datasets of pollen- and archaeology-based data of past LU and LC that can be used in palaeoclimate
modeling or the evaluation of DVMs and ALCCs (PAGES LandCover6k (Gaillard et al., 2015; Morrison et al.,
2016; Harrison et al., 2020)).
The only gridded pollen-based REVEALS reconstructions of plant cover for the purpose of climate modeling
published so far are those for NW-Central Europe North of the Alps (five time windows of the Holocene)
(Trondman et al., 2015) and entire Europe through the Holocene (11.7 ka BP to present) (Githumbi et al., 2022).
A comparison of Trondman et al. (2015) reconstruction with the ALCC scenarios from HYDE 3.1 (Klein
Goldewijk et al., 2011) and KK10 (Kaplan et al., 2009) suggests that the KK10-simulated deforestation is closer
to the REVEALS estimates of open land (OL) cover than the HYDE 3.1 deforestation (Kaplan et al., 2017). In a
study using a regional climate model (Strandberg et al., 2014), it was found that the effect on mean summer and
winter temperatures of anthropogenic deforestation equaling KK10-simulated deforested land in Europe between
6 and 0.2 ka BP varied between ca. -1 °C and +1 °C depending on the season and geographical location. This
indicates that LULCs in the past did matter in terms of climate change and was further confirmed in a recent
palaeoclimate modelling study of the climate at 6 ka BP using the latest pollen-based REVEALS reconstruction
of plant cover in Europe (Githumbi et al., 2022; Strandberg et al., 2022). Besides the gridded REVEALS
reconstructions at the continental scale of Europe mentioned above, gridded REVEALS reconstructions along N-



S and W-E transects through Europe between 11.7 ka BP and present were used to disentangle the effects of
climate and land-use change on Holocene vegetation (Marquer et al., 2017). Moreover, gridded maps of pollen-
based REVEALS estimatess of open land cover in the northern hemisphere (N of 40˚) were published for a couple
of Holocene time windows (Dawson et al., 2018).
Several reconstructions of the biomes (Ni et al., 2010, 2014) and vegetation cover (Tian et al., 2016) of China
during the Holocene are available. However, these reconstructions do not provide quantitative information on the
spatial extent of deforested land within woodland biomes or vegetation types including both trees and herbs.
Therefore, they are of limited value for use in palaeoclimate modelling or the evaluation of DVM-simulated
vegetation cover or ALCC scenarios.
The dataset of gridded pollen-based REVEALS estimates of plant cover for temperate and northern sub-tropical
China presented in this paper is based on the REVEALS estimates published in Li et al. (2020). It includes, for 25
consecutive time windows of the Holocene, cover estimates for 27 plant taxa, further grouped into estimates of
cover for six plant functional types (PFTs) and three land-cover types, i.e. coniferous tree (CT), broadleaved tree
(BT) and C3 herbs/open land (C3H/OL). PFTs are either single taxa (mainly genus, such as *Pinus*, *Betula*, etc.) or
groups of taxa. Here we briefly describe the methods used and their rationales,  present a selection of maps of the
cover of CT, BT and C3H/OL for eight time windows of the Holocene, and discuss the reliability and limitations
of the dataset as well as its current and potential uses. The entire dataset is available at
https://data.tpdc.ac.cn/en/disallow/d18d2b7e-25fe-49da-b1bd-2be6014162b0/.

**2 Data and methodology**
For the sake of consistency and comparison between regions and continents, and to fullfil the criteria required for
a contribution to the Past Global Changes (PAGES) LandCover6k working group (2015−2021;
https://pastglobalchanges.org/science/wg/former/landcover6k/intro), the application of the REVEALS model
follows the protocol used for the REVEALS reconstructions performed in Europe (Mazier et al., 2012; Trondman
et al., 2015) as closely as possible. For the full protocol of the REVEALS reconstructions for China, see Li et al.
119 (2020).

**2.1 Pollen data**
The pollen records selected for this study are from the pollen-data archive published by Cao et al. (2013) and from
individual contributors. The pollen-data archive includes over 230 pollen records for temperate and northern
subtropical China covering all or parts of the Holocene. However, only 94 pollen records met the criteria required
for a contribution to PAGES LandCover6k (Trondman et al., 2015; Githumbi et al., 2022): i.e. the pollen records
are from lake sediments and/or peat deposits in small to large basins, pollen identification is of good quality, their
chronology is based on $\geq$ 3 dates ($^{14}$C or other types of dates), and they have a temporal resolution of minimum
two pollen counts per 500 years. All chronologies were carefully examined. If required,  new age-depth models
were established using the BACON software (Blaauw and Christen, 2011). Hereafter, all ages are given in ka BP
(1000 years before present;  BP= 1950 CE).




The metadata table (Table S1) includes, for each pollen record/site, the vegetation zone, the number of the site
group (Gr; explanations below), the site name and its latitude, longitude and elevation, the province, the site (lake
or bog) area and calculated radius, the basin type (lake or bog), the type of pollen data (original raw pollen counts,
or calculated pollen counts using information from published pollen diagrams), the dating method and number of
dates, the timespan covered by the pollen record, the mean time resolution of the pollen counts,  and the literature
reference.

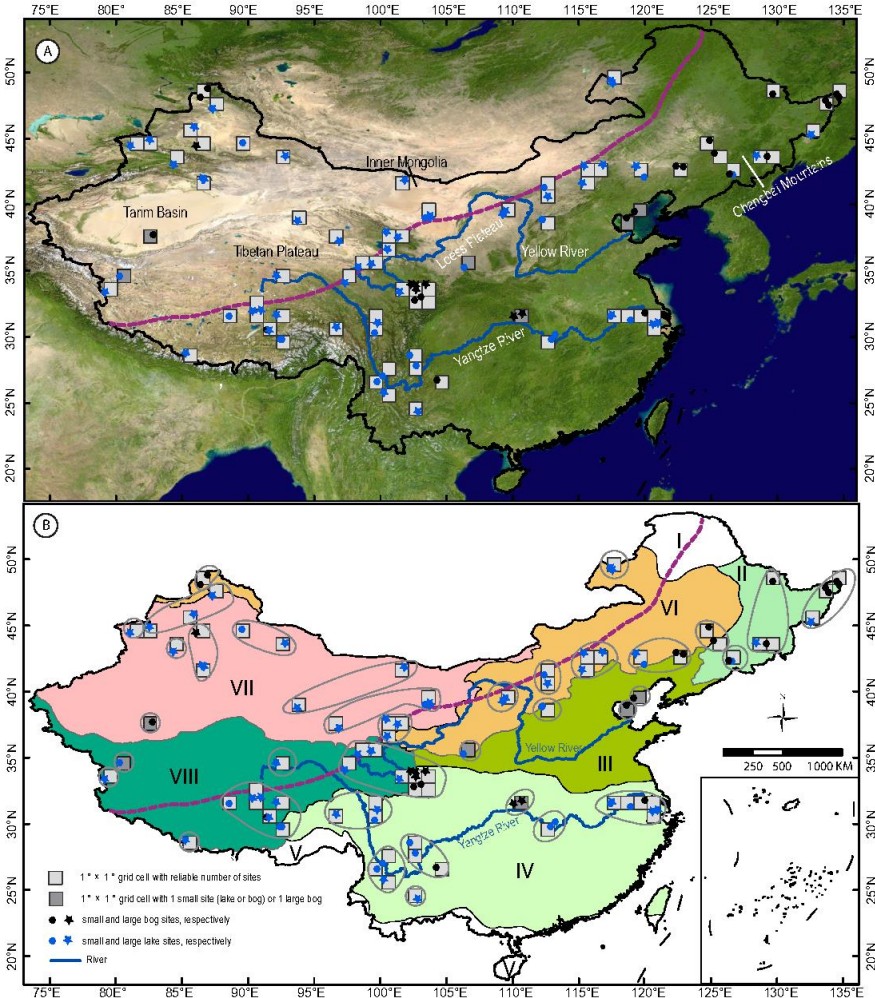


**Figure 1: Study region and selected Holocene pollen records. A. Satellite image from planetobserver showing the major**
**mountains, rivers and geographical regions mentioned in the text. B. Map from Li et al. (2020), modified: vegetation**
**zones in China following Hou (2019) and site groups delimited by a grey line. Grid cell reliability in terms of REVEALS**
**estimates of plant cover is indicated by light grey (high reliability) and dark grey (low reliability) depending on number**
**and type of pollen records (see text for detailed explanations). Roman numbers refer to vegetation zones: I. Boreal forest,**
**II. Coniferous-deciduous mixed forest, III. Temperate deciduous forest, IV. Subtropical broadleaved evergreen and**
**deciduous forest, V. Tropical monsoonal rainforest, VI. Temperate steppe, VII. Temperate desert, VIII. Highland**
**vegetation.**




**2.2 The REVEALS model and rationales for the model-application protocol**

A full description of the REVEALS model and its assumptions is published in Sugita (2007a). The model was developed to estimate plant cover at a regional scale using pollen data from large lakes. It is a modification of the R-Value model  (Davis, 1963) that corrects pollen percentage biases caused by inter-taxonomic differences in pollen productivity and dispersion. Empirical tests in southern Sweden and northern America suggest that pollen records from lakes ≥50 ha provide reliable pollen-based REVEALS estimates of regional plant cover (Hellman et al., 2008a,b; Sugita et al., 2010). The rationales behind the general protocol used for the gridded REVEALS reconstructions are presented in detail in Mazier et al. (2012), and Trondman et al. (2015). Major rationales are those motivating the use of a 1˚x1˚ spatial resolution (grid-cell size), a 500 years time resolution (except for the three most recent time windows), and all suitable pollen records from 1 large and small sites. The choice of the spatial scale is based on a test performed in southern Sweden demonstrating that REVEALS estimates of modern plant cover using pollen assemblages from surface lake sediments were in good agreement with the actual plant cover within areas of 50 km × 50 km and 100 km × 100 km (Hellman et al., 2008b). In addition, this spatial scale is appropriate for palaeoclimate modelling, either with global or regional climate models (e.g.Strandberg et al., 2014; 2022). The time resolution is motivated by the influence of the size of pollen counts on the size of the REVEALS estimates standard errors.  A time resolution  of 500 years ensures that a maximum of the REVEALS reconstructions have low SEs and it is still meaningful for the study of past land-cover changes over several millennia. As pollen counts are generally available at a higher time resolution for the last 1000 years, and because land-cover changes were often more rapid during the recent millennium than through the earlier millennia, the length of the three most recent time windows were fixed to 350, 250, and 100 years (0.7−0.35 ka BP, 0.35−0.1 ka BP, and 0.1 ka BP to present). The relevance and suitability of using pollen records from both large and small sites for REVEALS applications in order to increase the reliability of the pollen-based estimates of plant cover within each grid cell is confirmed by simulation tests in Sugita (2007a) and empirical tests in southern Sweden (Trondman et al., 2016) (see Li et al. (2020) for more details). In the absence of pollen records from large lakes, the larger the number of small sites (lakes or bogs), the better the REVEALS result. However, bogs (large and small) violate one of the assumptions of the REVEALS model, i.e. "no vegetation is growing on the deposition basin" (Sugita, 2007a). Violation of this assumption has been shown to bias REVEALS  results most significantly in the case of large bogs, while pollen records from multiple small bogs use to provide reliable estimates of plant cover (Mazier et al., 2012; Trondman et al., 2016).

In this gridded REVEALS reconstruction of plant cover in China, a deviation from the standard protocol used in Europe has been to perform the REVEALS reconstructions using pollen records within larger areas than a single 1˚×1˚ grid cell. Due to the low spatial density of the 94 selected pollen records in this study, the pollen records were grouped for the application of the REVEALS model within coherent regions with comparable biogeographical characteristics and similar vegetation histories (see Li et al. (2020) for details). It implies that, in these cases, several adjacent grid cells (2−8) have the same REVEALS estimates. The advantage is that the REVEALS estimates are more reliable and have lower SEs than they would have been if the reconstructions had been performed for the individual 1˚x1˚grid cells with only few pollen records. The pollen records within 57 of 75





1˚x1˚grid cells belong to such groups of pollen records (19 in total) from regions larger than a single grid cell. The
remaining 18 grid cells include one or two pollen records that could not be grouped with additional pollen records.

**2.3 Parameter settings, REVEALS runs and calculation of cover for groups of plant taxa**
Parameters needed to run the REVEALS model are relative pollen productivity estimates (RPPs) and their standard
deviation (SD), fall speed of pollen (FSP), maximum extent of regional vegetation ($Z_{max}$; km), wind speed (m/s),
and atmospheric conditions. We used the mean RPPs estimates with their related SDs and the FSPs of 27 plant
taxa from the synthesis of available RPP and FSP values in temperate China (Li et al., 2018b), a $Z_{max}$ of 100 km,
a wind speed of 3 m/s, and stable atmospheric conditions. Other parameters needed are the basin type (lake or bog)
and its size (radius in m). We applied two models of pollen dispersion and deposition, the "Prentice model"
(Prentice, 1985) for bogs and the "Prentice-Sugita" model (Sugita, 1993) for lakes.
Before running the REVEALS model, the pollen counts of the 27 plant taxa within each time window were
summed up in each pollen record. The REVEALS model runs and all calculations of mean REVEALS estimates
from several pollen records and for group of plant taxa were performed using three computer programs written by
Shinya Sugita (unpublished). The latest version of the REVEALS computer program, LRA.REVEALS.v6.2.4.exe
(Sugita, unpublished) and example files are available at the link https://1drv.ms/u/s!AkY-
0mVRwOaykdgmINfXVsC-4t4n5w?e=7U55hO. The REVEALS model was run separately with pollen records
from bogs (with the Prentice's model) and lakes (with the Prentice-Sugita model) for each group of pollen records .
These model runs result in two different mean REVEALS estimates (and their SEs) of cover for the 27 plant taxa,
one from bog(s) and one from lake(s). The final mean REVEALS estimates of cover for the 27 plants taxa (from
bog(s) + lake(s))are then calculated. The SEs of the final mean REVEALS estimates for each group of pollen
records are obtainedusing the delta method (Stuart and Ord, 1994) (see Li et al., 2020 for details).
For use in climate models and evaluation of HYDE, KK10, and DVMs (see Introduction), we also calculated the
mean REVEALS estimates (and their SEs) of cover for groups of taxa, i.e. plant functional types (PFTs) and land-
cover types (LCTs). To do so, the 27 plant were harmonized with six PFTs defined for China by Ni et al. (2010,
2004), and with the three LCTs CT, BT and C3H/OL (Table 1). Note that Li et al. (2020) used slightly different
PFTs where Cupressaceae, Poaceae, Cyperaceae and Rosaceae were treated as separate PFTs to make the
interpretation of changes in the amount of conifers and herbs in terms of regional versus local - and natural versus
anthropogenic - vegetation easier. Moreover, Rubiaceae and Elaeagnaceae were classified as belonging to the
temperate shade-tolerant broadleaved evergreen trees, and *Castanea* and *Juglans* were grouped with the herbs
(openland) and anthropogenic indicators (including planted trees). In this study we used the PFT classification
provided in Table 1 in which Cupressaceae is grouped with *Pinus* as belonging to PFT TeNE (temperate shade-
intolerant needle-leaved evergreen trees), Elaeagnaceae, *Castanea*, *Juglans* with broadleaved trees as belonging
to PFT TeBS (Temperate shade-tolerant broadleaved summer green trees), and Cyperaceae, Poaceae, Rosaceae,
and Rubiaceae with all herbs as belonging to PFT C3H/OL (C3 Herbs/openland). We propose that this
classification is more appropriate for use in climate modelling contexts than that used in Li et al. (2020) in which





the major aim of the study was to interpret the pollen-based plant-cover reconstruction in terms of regional
vegetation cover.
For more details on parameter setting, REVEALS runs, models of pollen dispersion and deposition, and the delta
method, the reader is referred to Li et al. (2020).
Table 1: Aggregation of pollen morphological types into Land-cover types (LCTs) and plant functional types (PFTs) (following
Ni et al., 2010, 2014). Fall speed of pollen (FSP) and mean relative pollen productivities (RPPs) with standard deviation (SD)
in brackets (dataset Alt2 of Li et al., 2018b). The number of values available in the calculation of the mean RPPs and location
of the RPP studies in terms of vegetation zones are also provided. Roman numbers refer to the vegetation zones: I. Boreal
forest, II. Coniferous-deciduous mixed forest, III. Temperate deciduous forest, IV. Subtropical broadleaved evergreen and
deciduous forest, V. Tropical monsoonal rainforest, VI. Temperate steppe, VII. Temperate desert, VIII. Highland vegetation.

| Land cover types | PFTs | PFTs definition | Plant taxa/Pollen-morphological types | FSP(m/s) | RPP(SD) | Number of RPPs | Location of RPP studies (Vegetation zones) |
|---|---|---|---|---|---|---|---|
| Coniferous Tree | TeNE | Temperate shade-intolerant needle-leaved evergreen trees | *Pinus* | 0.035 | 18.37(0.48) | 4 | II, III, |
| | | | Cupressaceae | 0.010 | 1.11(0.09) | 1 | III |
| | BNS | Boreal needle-leaved summer green trees | *Larix* | 0.126 | 2.14(0.24) | 3 | II, III |
| Broadleaved Trees | IBS | Boreal shade-intolerant broadleaved summer green trees | *Betula* | 0.014 | 12.42(0.12) | 3 | II, III |
| | TeBS | Temperate shade-tolerant broadleaved summer green trees | *Castanea* | 0.004 | 11.49(0.49) | 1 | III |
| | | | Elaeagnaceae | 0.012 | 8.88(1.30) | 1 | III |
| | | | *Fraxinus* | 0.017 | 3.94(0.73) | 1 | II |
| | | | *Juglans* | 0.031 | 7.69(0.24) | 1 | III |
| | | | *Quercus* | 0.019 | 5.19(0.07) | 3 | II, III |
| | | | *Tilia* | 0.028 | 0.65(0.11) | 1 | II |
| | | | *Ulmus* | 0.021 | 4.13(0.92) | 2 | II,III |
| | TeBE | Temperate shade-tolerant broadleaved evergreen trees | *Castanopsis* | 0.004 | 11.49(0.49) | 1 | III |
| | | | *Cyclobalanopsis* | 0.019 | 5.19(0.07) | 3 | II,III |
| Openland | C3H | C3 Herbs | Amaranth./Chenop. | 0.013 | 4.46(0.68) | 2 | VI, VIII |
| | | | *Artemisia* | 0.010 | 21.15(0.56) | 4 | II, VI |
| | | | Asteraceae | 0.019 | 4.4(0.29) | 2 | VI |
| | | | Brassicaceae | 0.012 | 0.89(0.18) | 1 | III |
| | | | *Cannabis/Humulus* | 0.010 | 16.43(1.00) | 1 | III |
| | | | Convolvulaceae | 0.043 | 0.18(0.03) | 1 | VI |
| | | | Cyperaceae | 0.022 | 0.44(0.04) | 2 | III, VIII |
| | | | Fabaceae | 0.017 | 0.49(0.05) | 2 | III, VI, |
| | | | Lamiaceae | 0.015 | 1.24(0.19) | 2 | VI |
| | | | Liliaceae | 0.013 | 1.49(0.11) | 1 | VI |
| | | | Poaceae | 0.021 | 1(0) | 6 | II, III, V, VI, VIII |
| | | | Ranunculaceae | 0.007 | 7.77(1.56) | 1 | II |
| | | | Rosaceae | 0.009 | 0.22(0.09) | 1 | VI |
| | | | Rubiaceae | 0.010 | 1.23(0.36) | 1 | III |


## 2.4 Data format

The dataset of pollen-based REVEALS estimates of Holocene plant cover for temperate and northern sub-tropical
China comprises four csv files with the REVEALS proportions of plant cover (and related SEs) in 75 1˚x 1˚grid



cells and 25 time windows for 27 taxa (Data1.plants.csv), six PFTs (Data2.6PFTs.csv) (PFT classification as in
Table 1), three land-cover types (Data3.LCTs.csv) and ten PFTs (Data4.10PFTs.csv) (PFT classification as in Li
et al. (2020)). Two additional files are complementing the REVEALS dataset, the metadata file (Table S1) (see
section 2.1 pollen data for details) and a table providing details on the number and types of sites used in the
REVEALS reconstruction for each grid cell and each time window (Table S2). The REVEALS excel data files
and Tables S1 and S2 (also in Supplementary Material) are available at
https://data.tpdc.ac.cn/en/disallow/d18d2b7e-25fe-49da-b1bd-2be6014162b0/.

**3. Results**

As an illustration, we describe below maps of the REVEALS reconstructed cover for the three land-cover types
CT, BT and C3H/OL for eight selected time windows of the Holocene that provide snap shots in time of
significantly different composition of land-cover types between 11.7 ka BP and present. For each land-cover type
the maps are described from the oldest (11.7–11.2 ka BP) to the youngest (0.1 ka–present) and each map in
comparison to the former one, e.g. for the 9.7–10.2 ka BP map changes are expressed in comparison to the 11.7–
11.2 ka BP map. The descriptions start with information extracted from Li et al. (2020) on the modern occurrence
and Holocene history (in terms of pollen-based REVEALS cover) of the taxa constituent of the land-cover type in
question.

**3.1 Open Land (C3H/OL; Figure 2)**

OL is the sum of the reconstructed cover of 14 herb taxa for which RPPs are available. Poaceae, Cyperaceae,
Amaranthaceae/Chenopodiaceae and *Artemisia* are often represented by high pollen percentages during the
Holocene. Other herbs that can be relatively well represented during most of the Holocene are Asteraceae,
Brassicaceae, Ranunculaceae, Rosaceae, and Rubiaceae. Pollen from Convolvulaceae, Fabaceae, Lamiaceae and
Liliaceae can be quite common over some periods of the Holocene, while *Cannabis/Humulus* is not frequent.
These herbs characterize today primarily open vegetation, i.e. temperate xerophytic shrubland and grassland, desert,
and tundra, as well as human-induced vegetation (cultivated and grazing land). The REVEALS reconstructions
suggest that the cover of Poaceae, Cyperaceae and Rosaceae during the Holocene is often equal or larger than the
cover of all remaining 11 herbs together, although *Artemisia* and Amaranthaceae/Chenopodiaceae can also reach
a relatively large cover (Li et al., 2020).
The time window 11.7−11.2 ka BP is characterized by OL cover values >80% in most grid cells of northwestern
China and the Tibetan Plateau. A small number of grid cells have OL values of 40−60% or 60−80% in
southwestern China and Inner Mongolia. A few grid cells have OL values 40−60% in the lower reach of the
Yangtze River region, and around 20−40% or 10−20% in northeastern China. The time window 10.2−9.7 ka BP
shows an increase in OL cover of 10% in the grid cells of northeastern China, and an increase to 60−80% or > 80%
in some grid cells of Inner Mongolia and the lower reach of the Yangtze River basin, while a decrease of 20% is
seen in a few grid cells in southwestern China. At 8.2−7.7 ka BP, the OL cover declines in most of the grid cells,
in particular in one grid cell of the Loess Plateau, three grid cells of central Inner Mongolia, and five grid cells of
the lower reach of the Yangtze River region, where OL decreased by 20−40%, whilst a decrease of 10−20% is

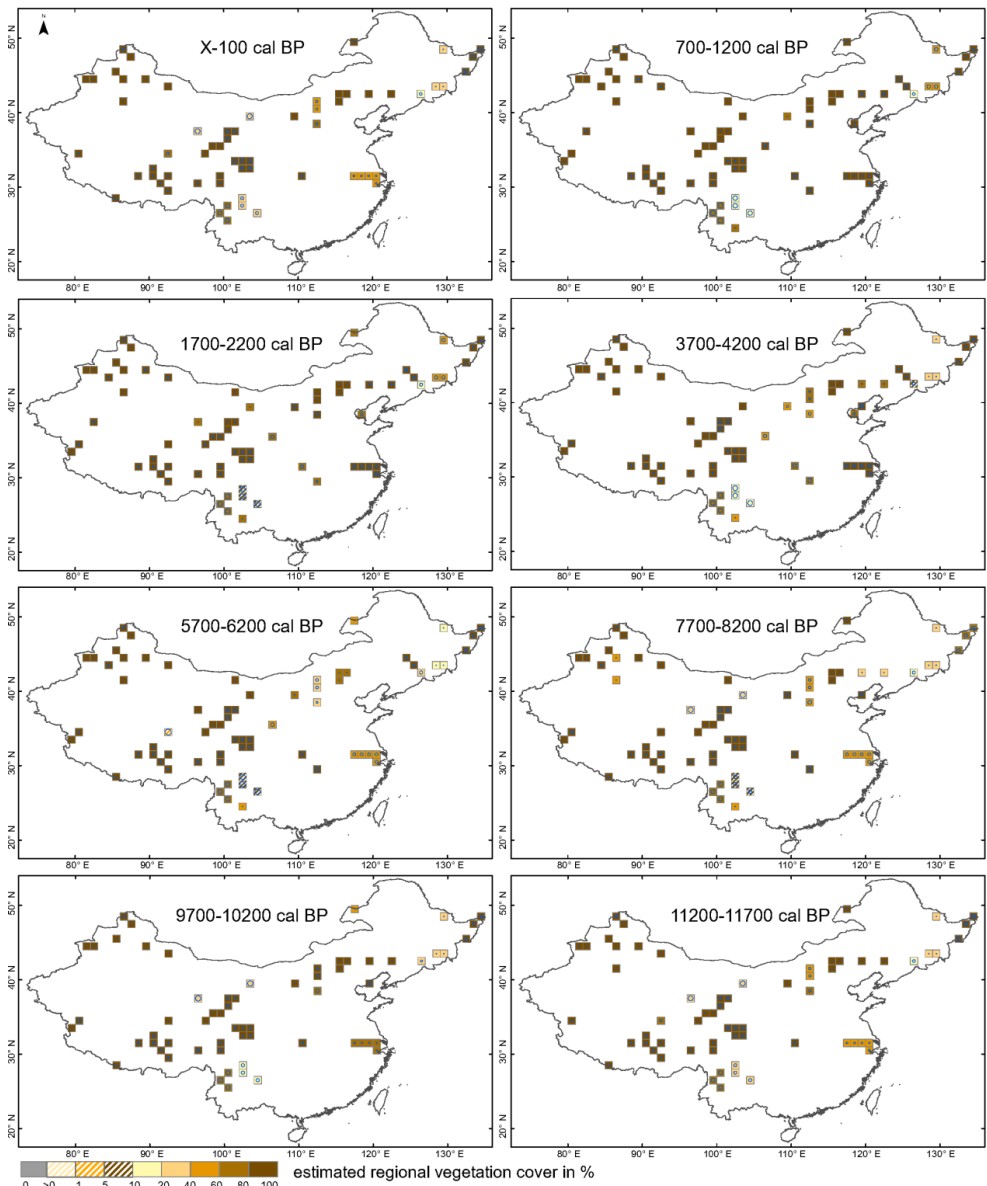

**Figure 2. Grid-based REVEALS estimates of C3 herbs/Open Land (C3H/OL) cover for eight selected time windows of the Holocene.** Percentage cover in intervals of 1% (>0−1%), 4% (>1−5%), 5% (>5−10%), 10% (>10−20%), and 20% (>20−100%) represented by increasingly darker colours from >0−1% to >5−10% and from >10−20% to 80−100%. Grid cells without pollen data for the time window, but with pollen data in other time windows are shown in grey. Uncertainties on the REVEALS estimates are illustrated by blue circles of various sizes corresponding to the coefficient of variation (standard error (SE) divided by the grid cell mean REVEALS estimate (RE)). If SE ≥ RE, the blue circle fills the entire grid cell. SE ≥ RE also implies that RE is not different from zero, which is the case primarily for low RE values.



seen in three grid cells of northwestern China. At 6.2−5.7 ka BP, OL shows a further decrease of 20% in Inner
Mongolia, three grid cells in northeastern China, and several grid cells in southwestern China, and a decrease of
60% in one grid cell of the central Tibetan Plateau, whilst an increase of 40% is observed in two grid cells of
northwestern China. At 4.2−3.7 ka BP, OL cover is > 80% in most of the regions of northwestern China. An
increase of 50% is observed in the lower reach of the Yangtze River region and the grid cells of Inner Mongolia,
of 20% in one of the grid cells of the Loess Plateau and three grid cells of southwestern China, and of 10−20% in
three grid cells of northeastern China. In contrast, a decrease of OL cover of 10−30% is seen in one grid cell of
northeastern China. At 2.2−1.7 ka BP, OL cover has increased in almost all regions except for a decrease of 20%
in southwestern China. The increase of OL cover is of 40% in Inner Mongolia and Shanxi, and 20% in three grid
cells in the Changbai Moutain region. Over the last ca. 100 years  (0.1 ka BP−present), there is no major change
in OL cover, except an increase of 10% in southwestern China and a decrease of 20% in several grid cells of
northeastern China.
3.2 Coniferous Trees (CT;  Figure 3)
CT is the sum of the reconstructed cover of three conifer taxa for which RPPs are available, *Pinus* and
Cupressaceae (PFT TeNe) and *Larix* (PFT BNS) (Table 1). We chose to use only RPP values estimated in China
(RPP synthesis of Li et al. (2018b)) and, therefore, did not produce REVEALS estimates of the cover of *Abies* and
*Picea* (Li et al., 2020). Today, these two taxa are common together with *Pinus* and *Larix* in the boreal forests and
coniferous-broadleaved mixed woodlands (zones I and II, respectively). *Abies* and *Picea* also form woodland
patches in the westernmost part of  the subtropical broadleaved evergreen and deciduous forest (zone IV), and
*Abies* and *Pinus* characterize the woodlands of the  zone IV southwestern part. Of the three conifer taxa for which
REVEALS  reconstructions are available, *Pinus* is the one with significant cover over most of the Holocene in all
regions characterized by coniferous woodland (or woodland patches) today in central and eastern-northeastern
China (Li et al., 2020). *Pinus* has a relatively large cover throughout the Holocene in zone IV southwestern part,
zone VI western part and zone II central part, while it has lower cover in zone IV eastern part. Some cover of *Pinus*
has some cover from 7 ka BP in zone VI eastern part and relatively high cover from 4.5 ka BP in zone II
southeastern part and zone III eastern part. A significant cover of Cupressaceae was reconstructed for the early
Holocene from some pollen records in zone IV western part and zone VII easternmost part (temperate desert), and
for most of the Holocene in zone VI western and northernmost parts (temperate steppe) (Li et al. 2020). *Larix* is
represented in zones II and VI central and northernmost parts either by continuous high cover throughout the
Holocene alternatively the Late Holocene only, or by scattered occurrences of high cover through time (Li et al.,

310    2020).

There is a consistent increase in CT cover in most grid cells over northern China during the first half of the
Holocene with maximum values sometime between 8 and 5 ka BP (the timing depending of the region), before a
steady decline of the values of CT cover. The time window 11.7−11.2 ka BP is characterized by CT cover values
of over 80% in one grid cell of northeastern China, 10−20% or 20−40% in southwestern China, and 10−20% in
two grid cells of the eastern part of northwestern China and in the lower reach of the Yangtze River region.
Elsewhere CT cover is lower than 10%. At 10.2–9.7 ka BP, the CT cover values have decreased in almost all grid



**Figure 3. Grid-based REVEALS estimates of Coniferous Trees (CT) cover for eight selected time windows of the Holocene. Percentage cover in intervals of 1% (>0−1%), 4% (>1−5%), 5% (>5−10%), 10% (>10−20%), and 20% (>20−100%) represented by increasingly darker colours from >0−1% to >5−10% and from >10−20% to 80−100%. Grid cells without pollen data for the time window, but with pollen data in other time windows are shown in grey. Uncertainties on the REVEALS estimates are illustrated by blue circles of various sizes corresponding to the coefficient of variation (standard error (SE) divided by the grid cell mean REVEALS estimate (RE)). If SE ≥ RE, the blue circle fills the entire grid cell. SE ≥ RE also implies that RE is not different from zero, which is the case primarily for low RE values.**






cells, with a decline of 10% in the lower reach of the Yangtze river, and 10−20% or 20−40% in a few grid cells
of northwestern China. CT cover is slightly higher in the 8.2−7.7 ka BP time window in most grid cells of
northeastern China (10−20%), while a small drop is seen in the grid cells of the western part of southwestern China
and eastern part of northwestern China. The time window 6.2−5.7 ka BP is characterized by a decrease of CT
cover with 10−20% in northeastern China and 40% or 60% in a few grid cells of northwestern China. In contrast,
CT cover has increased with 20−40% and ca 5% in Inner Mongolia and southwestern China, respectively. From
4.2−3.7 ka BP, CT cover exhibits a further decrease with maximum 20% in most grid cells of Inner Mongolia and
southwestern China. The CT cover at 2.2−1.7 ka BP is even lower, with a decline of 10% and >10−20% in the
eastern part of northwestern China and the western part of northeastern China, respectively. There is however a
slight increase in CT cover with 2% in northwestern China and the lower reach of the Yangtze River. At 1.2−0.7
ka BP, the CT cover has decreased with 2% on the Tibetan Plateau, in northwestern China, and the lower reach of
the Yangtze River. An increase in CT cover with ca. 10% during the last century (0.1 ka BP−present) is found in
southwestern, eastern, and most of northeastern China, while a decrease is seen in some grid cells of northeastern
China.
**3.3 Broadleaved Trees (BT; Figure 4)**
BT is the sum of the reconstructed cover of ten broadleaved tree taxa for which RPPs are available, *Betula* (PFT
IBS), *Castanea*, *Eleagnaceae*, *Fraxinus*, *Juglans*, *Quercus*, *Tilia*, and *Ulmus* (PFT TeBS), *Castanopsis* and
*Cyclobalanopsis* (PFT TeBE) (Table 1). *Betula* has a significant cover throughout the Holocene in zone II and
most of zone IV (Li et al. 2020). The summer-green broadleaved tree taxa (TeBS) are characteristic of zones II,
III and IV with relatively large cover throughout the Holocene, and of the southern boarder of vegetation zone VI
with large cover in particular through Mid Holocene. The evergreen broadleaved tree taxa *Castanopsis* and
*Cyclobalanopsis* are characteristic of vegetation zone IV with relatively large cover in most of the zone (Li et al.,

349  2020).

The Holocene changes in cover of BT show similar trends as those for CT, with a steady increase during the first
half of the Holocene with the highest values found in the time windows from 8.2−7.7 ka BP to 5.2−4.7 ka BP
(depending on the region) followed by a steady decrease through the Late Holocene. The oldest time window
11.7–11.2 ka BP is characterized by the largest BT cover of the Holocene (>80%) in several grid cells of
northeastern China, and the second largest BT cover (20−40%) in some grid cells of Inner Mongolia and the lower
reach of the Yangtze River region. In contrast, BT cover is <2% in northwestern China and on the Tibetan Plateau.
At 10.2−9.7 ka BP, BT cover has increased with ca. 10% in a few grid cells of northeastern China, while it has
decreased with 10% in two grid cells of Inner Mongolia. An increase of BT cover with 10% or 20% in time window
8.2−7.7 ka BP is seen in some grid cells of northeastern China and the Yangtze River lower reach, while there is
a decrease with 5% in the three grid cells of northeastern China. At 6.2−5.7 ka BP, BT cover has decreased with
20% in two grid cells of central Inner Mongolia and three grid cells of southwestern

**Figure 4. Grid-based REVEALS estimates of Broadleaved Trees (BT) cover for eight selected time windows of the Holocene. Percentage cover in intervals of 1% (>0−1%), 4% (>1−5%), 5% (>5−10%), 10% (>10−20%), and 20% (>20−100%) represented by increasingly darker colours from >0−1% to >5−10% and from >10−20% to 80−100%. Grid cells without pollen data for the time window, but with pollen data in other time windows are shown in grey. Uncertainties on the REVEALS estimates are illustrated by blue circles of various sizes corresponding to the coefficient of variation (standard error (SE) divided by the grid cell mean REVEALS estimate (RE)). If SE ≥ RE, the blue circle fills the entire grid cell. SE ≥ RE also implies that RE is not different from zero, which is the case primarily for low RE values.**



China. A further decrease of cover has occurred 4.2−3.7 ka BP, with 20−30% in the lower reach of the Yangtze
river and northeastern China, and with 10% in Inner Mongolia. BT cover has further decreased at 2.2−1.7 ka BP
with 10% in the western and central part of northeastern China, and at 1.2−0.7 ka BP with < 10%, 10%, or 20% in
northeastern China and the the Yangtze River lower reach. In the last century (0.1 ka BP−present), BT cover has
increased with 30% and 20% in the eastern part of northeastern China and in southwestern China, respectively. In
contrast, the three grid cells in the western part of northeastern China are characterized by a strong decrease in BT
cover.
**4. Reliability and limitations of the dataset**
**4.1 Accuracy and reliability of the REVEALS estimates of plant cover**
For a detailed description of the accuracy and reliability of the REVEALS reconstructions, the reader is referred
to Li et al. (2020). The quality of the REVEALS reconstructions are mainly reliant on input data (pollen counts
quality and size), the reliability of pollen records chronologies and relative pollen productivities used (RPPs), the
type and size of the pollen records sites (lakes or bogs), the number of pollen records used for reconstruction in
each grid cell, and variation between pollen counts within a grid cell. The standard errors (SEs) of the REVEALS
estimates are a measure of their accuracy and reliability. They are based on the SDs of RPPs and the between-site
variation in pollen assemblages within a grid cell (or several adjacent grid cells in the case of the reconstruction
presented here, see methods for more details). If SE < mean REVEALS estimate of cover, the result is considered
to be reliable, which is the case for over 85% of the reconstructions. If SE ≥ mean REVEALS estimates of cover,
the result is not different from zero and, therefore, not reliable. The latter occurs mainly in the lower reach of the
Yangtze River region.
Other issues may influence the reliability of the REVEALS estimates of plant cover. REVEALS was intended for
pollen records in large lakes (Sugita, 2007a). Pollen records from bogs violate the assumption of the model that
no plants are growing on the surface of the deposition basin. Therefore, pollen-based REVEALS estimates from
bogs may be biased by local cover of major plant taxa such as Poaceae and Cyperaceae, in particular if bogs are
large. The problem is discussed in detail in Li et al. (2020), where the cover of openland was considered to be
overestimated in some grid cells due to this phenomenon, in particular in northeastern China. This issue and the
theoretically inadequate application of REVEALS using a single pollen record from a small site (lake or bog) or a
large bog in a grid cell are indicated as providing less reliable or unreliable REVEALS reconstructions of plant
cover in Figure 1 (dark grey grid cells). Moreover, the number of sites and their type (lake or bog) and size (large
or small) are provided for each site group (grid cell) and time window in Table S2. Uncertainty related to the RPPs
used is another factor influencing reliability of the REVEALS reconstructions. We use the mean RPPs from the
Chinese synthesis published in Li et al (2018b). The assumptions are that RPPs are constant through time and the
mean RPPs are a good approximation for the plant taxa over the entire study region. Although we do not know
whether RPP was constant through the Holocene for the plant taxa used in the reconstructions, the assumption is
necessary if we are to reconstruct changes in the abundance or absolute cover of plants from changes in pollen
percentages over time (e.g.Birks and Birks, 1980; Sugita, 2007a). Mean RPPs are most reliable for large regions
if they are based on a large number of RPP values that are well distributed within the study region, and if these
values do not differ very significantly from each other. A measure of variability among RPP values is provided by



the SD of the mean RPP, which is in turn imbedded in the REVEALS estimate's SE of a plant taxon's cover.
However, none of the SDs is very large in relation to the mean RPP values we are using (Table 1). SD is larger
than a tenth of the mean RPP value for ten taxa of the 27 taxa used (i.e. Elaeagnaceae, *Fraxinus*, *Tilia*, *Ulmus*,
Amaranthaceae/Chenopodiaceae, Brassicaceae, Convolvulaceae and Ranunculaceae; Table 1), however with SD
less than a fifth of the mean RPP value except for *Fraxinus*, *Ulmus*, Brassicaceae, and Ranunculaceae (SD ca. a
fifth of mean RPP), Rosaceae (SD ca. a third of mean RPP) and Rubiaceae (SD ca. a fourth of mean RPP).
Therefore, large SEs are probably seldom due to the RPPs' SDs. However, there is no way to measure the
uncertainty that may be caused by the use of a mean RPP value based on too few RPP values, or RPP values that
are not representative of all major vegetation zones of the study region. The number of values available in the
calculation of the mean RPPs and location of the RPP studies in terms of vegetation zones are provided in Table
1. This information provides a mean to identify RPPs that might be uncertain for REVEALS land-cover
reconstructions in general, or in particular for certain regions.There is only one RPP value for 14 of the 27 taxa in
this study, i.e. Cupressaceae, *Castanea*, Elaeagnaceae, *Fraxinus*, *Juglans*, *Tilia*, *Castanopsis*, Brassicaceae,
*Cannabis/Humulus*, Convolvulaceae, Liliaceae, Ranunculaceae, Rosaceae, and Rubiaceae. The REVEALS
estimates for these taxa should, therefore, be considered with caution. The REVEALS estimates for *Castanopsis*
and *Cyclobalanopsis* are also uncertain because, in the absence of RPPs for these two taxa, we used instead the
RPPs of *Castanea* and *Quercus*, respectively, assuming comparable pollen productivities  (see Li et al., 2020 for
further details on this issue).

**4.2 Limitations of the pollen-based REVEALS plant cover**

The REVEALS model estimates the proportion of each plant taxon in relation to the total cover of all taxa with
RPPs available (in this case 27 taxa) rather than its actual cover if all existing taxa could be considered. The same
consideration is valid for the REVEALS cover of the three major land-cover types C3H/OL, CT and BT. This is a
serious caveat if the taxa for which no RPP values are available represent a significant part of the pollen
assemblages. In this first dataset of REVEALS land-cover estimates, our decision to use exclusively Chinese RPPs
and, therefore, not reconstruct the cover of *Abies* and *Picea* is a major issue. This  may bias the results in
overestimating the cover of C3H/OL in particular, but also of BT. The latter needs to be kept in mind  in the
interpretation and use of the dataset for regions where *Abies* and *Picea* were common during part of, or the entire
Holocene, which was the case mainly in vegetation zones II and IV (see Results for CT for more details).
Another important caveat of all REVEALS reconstructions is that the cover of bareground in a landscape cannot
be inferred by the model. However, bareground was (and still is) a significant portion of the land cover in regions
characterized by desert, steppe, and high altitude vegetation (zones VI, VII and VIII in this study). So far, there is
only one attempt at estimating bareground in the past (Sun et al., 2022). It uses the modern relationship between
tree pollen and the cover of bareground in northern-central China, and the Modern Analog Technique (MAT) to
estimate the  the past cover of bareground using fossil pollen records from the same region. The MAT-estimated
cover of bareground is then used to correct REVEALS-estimated plant cover from the same fossil pollen records.
The results suggest that bareground covered 40 to 60% of the land and that the uncorrected REVEALS
reconstructions overestimate the cover of trees by ca. 50%, which can have implications if pollen-based REVEALS
land cover is used in palaeoclimate model experiments. In the context of palaeoclimate modelling, the





interpretation of the openland fraction (with or without bareground) in terms of deforestation (human-induced
decrease in tree cover) remains problematic due to the possible occurrence of herb taxa in both natural, climate-
induced and human-induced vegetation types, i.e. the reconstructed openland cover can be either natural or human-
induced, or both. This issue is discussed thoroughly in Li et al. (2020) as well as the difficulty to infer the
occurrence of past crops such as rice and millet from pollen records. Although pollen of cereals such as *Triticum*
(wheat), *Hordeum* (barley) and *Zea mays* (corn) can be separated from pollen of wild grasses, a RPP value for
these types of cereals could not be estimated in the study of Li et al. (2018b). Moreover, pollen grains from several
crops belonging to the families Fabaceae, Brassicaceae, Asteraceae, and Apiaceae cannot be separated from the
wild species (Ni et al., 2014).  Based on the considerations above, the interpretation of past changes in openland
cover needs to be cautious and The is a limitation of the gridded REVEALS land-cover dataset if used for
validation of ALCC scenarios and studies of human-induced land-cover change as a climate forcing.
Overestimation of deforestation in ALCCs can be detected in a comparison with REVEALS estimates of past
openland, whereas an underestimation cannot be demonstrated (Harrison et al., 2020). This issue is particularly
problematic in regions of northern China where steppes, desert, and meadows were dominant over most of the
Holocene. Similar limitations exist for the gridded REVEALS land-cover datasets in Europe, although less serious
as early agriculture developed primarily on land where woodland was the natural climate-induced vegetation cover
and only a smaller fraction of the continent was characterized by steppe vegetation (Trondman et al., 2015;
Githumbi et al., 2022; Strandberg et al., 2022).
The time resolution of the REVEALS reconstructions (500 years over most of the Holocene) is another limitation
in terms of quantification of land-cover change. A relatively low time resolution implies that major but rapid land-
cover changes will be missed or underestimated as they will be agglomerated into a mean cover over 500 years.
The chosen time resolution is a compromise to improve the quality of the REVEALS estimates rby increasing
pollen sums for pollen records characterized by a low time resolution of pollen counts (i.e. decrease the standard
error of the reconstruction, see methods for more details). Increasing the time resolution would be an advantage
only for regions, and periods of the Holocene, for which most pollen records have a high time resolution.
Finally, half of the REVEALS reconstructions (18 per time windows) are based on pollen records located within
several adjacent 1˚× 1˚ grid cells (a total of 58 1˚× 1˚ grid cells divided into 18 groups of 2 to 5 grid cells;  Figure
1) rather than within single 1˚× 1˚ grid cells (17 REVEALS reconstructions per time window).  This implies that
these 18 REVEALS estimates of cover (covering 58 1˚× 1˚ grid cells) represent a mean cover for areas of 1˚x 2˚
to 1˚× 5˚. The latter can be a limitation if the dataset of past land cover is used for studies in which the variability
of plant cover at a 1˚× 1˚ spatial scale is of importance. We opted for this deviation from the standard protocol
used in the REVEALS land-cover reconstructions for Europe (Trondman et al., 2015; Githumbi et al., 2022)
because of the low spatial density of pollen records in many parts of China and its negative consequence for the
quality of the REVEALS reconstructions if they were performed at a 1˚× 1˚ spatial scale implying a too low
number of pollen records per grid cell.

**5.  Potential application of the REVEALS estimates**



Quantitative reconstruction of land cover at regional to global scales is necessary for the study of climate-land
cover interactions using both regional and global climate models, and for evaluation of ALCC scenarios and
dynamic vegetation models. This first dataset of REVEALS land cover for temperate and northern subtropical
China is a contribution to PAGES LandCover6k, whose purpose is to provide datasets of Holocene pollen-based
land cover and archaeology-based land-use for (palaeo-)climate modeling (Gaillard et al., 2018; Harrison et al.,
2020). Such datasets are an alternative to reconstructions of vegetation cover using biomization  (Prentice and
Webb Iii, 1998) or the Modern Analog Technique (Overpeck et al., 1985). REVEALS reconstructions have the
advantage to provide estimates of cover for individual plant taxa that can be aggregated into cover of groups of
taxa such as PFTs or land-cover units. They can be used for various purposes, such as the evaluation of scenarios
of past deforestation (HYDE and KK) (Kaplan et al., 2017) and comparison with simulations of past vegetation
cover using dynamic vegetation models (Marquer et al., 2014, 2017), or climate modeling experiments looking
into past human-induced land coverl as a climate forcing (Strandberg et al., 2014; 2022). Such studies have not
been performed in China so far, although comparison of the REVEALS reconstructions of openland, CT and BT
cover presented here with HYDE 3.2 and KK10 is in progress. Further, studies attempting to disentangle the effects
of climate and land-use change on plant cover through the Holocene or looking into changes in diversity indexes
based on REVEALS estimates of past plant cover(e.g. studies by Marquer et al. (2014, 2017) in Europe), would
also be of great interest in a Chinese context. Another possible use of Holocene REVEALS-estimated of plant
cover is the comparison of regional plant-cover change with archaeological data to study the effect of large-scale
changes in population growth and settlement patterns and densityon vegetation cover in the past. A first attempt
at such a comparison in eastern China shows that phases of deforestation as interpreted from the REVEALS
estimates of open land cover between 6 and 3 ka BP are well correlated with changes in settlement densities over
the same time period, as suggested by archaeological data and population growth based on [14]C dates of
archaeological artefacts (Li et al., 2018a)

**6. Data availability**
All data files are available for public download at the National Tibetan Plateau Data Center (TPDC; Li et al., 2022;
https://data.tpdc.ac.cn/en/disallow/d18d2b7e-25fe-49da-b1bd-2be6014162b0/.). For more details on the files
available at the link, see section 2.4 on data format.
**7. Conclusions**
This paper describes the first datset of Holocene gridded pollen-based REVEALS reconstructions of plant taxa at
a $1° \times 1°$ spatial scale and continuous temporal scale of 500 years (350, 250, and 100 + x years from 0.7 k BP to
1950 CE + x years (x years is the number of years between 1950 CE and the year of coring) . The reconstructions
are based on 94 pollen records in temperate and northern subtropical China and include land-cover estimates for
27 plant taxa and aggregation to plant functional types and three land-cover types. The REVEALS model
assumptions and the limitations of this particular application are clearly stated,  in order to facilitate a correct and
cautious interpretation and assessment of the results. In particular, the consequences of the lack of estimates for
the cover of two major conifer trees (*Abies* and *Picea*), bareground, and crop land need to be taken into account in
any studies using the dataset, in particular for the vegetation zones II and IV (*Abies, Picea*), and VI, VII, and VIII
(bareground, crop land). Examples of uses are the evaluation of model-simulated vegetation cover and



deforestation from dynamic vegetation models and ALCC scenarios, respectively, as well as studies of past land-
use change as climate forcing during the Holocene. In all uses of the presented gridded REVEALS land-cover
dataset, the limitations of the REVEALS reconstructions have to be taken into account carefully (see Discussion
section for more details). Reconstructions of plant cover at a local spatial scale can be of value in archaeological
contexts. One of the input data required for the application of the LOcal Vegetation Estimates model (LOVE;
Sugita, 2007b) to estimate local plant cover is that regional plant cover. The dataset of gridded REVEALS
reconstructions may be a way to achieve reconstructions of local plant cover, with the condition that the pollen
records used for the LOVE application are not used in the REVEALS reconstructions of the dataset (Cui et al.,
2013; Mazier et al., 2015).
This dataset is the first generation of gridded REVEALS Holocene land-cover reconstructions for China. We
expect that, in the future, new generations of such datasets will develop, in which the quality and spatial extent of
the REVEALS estimates will be further improved, as more pollen records will be available, and additional RPP
studies will gradually increase both the number of RPP values per taxon and the number of taxa for which RPPs
are available.

**Author Contribution**
FL and MJG conceptualized and coordinated the study as a contribution to the PAGES working group
"LandCover6k". SS solved all specific issues related to the application of REVEALS in the context of China's
vegetation history and available pollen records . FL, XC, UH, and JN were responsible for collection of new pollen
records from individual authors. YZ contributed several published and unpublished pollen records and made
comments and edits to the manuscript. JN, CA, XH, YL, HL, AS, YY contributed pollen data. FL had the major
responsibility of pollen data files handling,and collection of related metadata,  and performed the REVEALS
application. FL and MJG  are responsible of thepaper's main objective and structure, FL prepared the first draft of
the manuscript, all figures and Tables, and finalization of the manuscript for submission. MJG contributed to text
in all its versions and  checked the final manuscript for content and English language. All co-authors contributed
with comments and corrections to  the manuscript.

**Competing interests**
The authors declare that they have no conflict of interest.

**Funds**
This work is supported by the National Science Foundation of China (NSCF) (PI Furong Li) [grant number,
42101143] and funds from the Swedish Strategical Research Area ModElling the Regional and Global Ecosystem,
MERGE (http://www.merge.lu.se/) (Furong Li (until 2019) and Marie-José Gaillard). We are also grateful for the
financial support from the Swedish Foundation for International Cooperation in Research and Higher Education
(STINT) and the NSFC [grant number, 41611130050] for a Sweden-China Exchange Grant 2016−2019 (PIs
Marie-José Gaillard). Furong Li (until 2020) and Marie-José Gaillard are grateful for support from the Faculty of
Health and Life Sciences at Linnaeus University, Kalmar, Sweden. This study was undertaken as part of the Past



Global Changes (PAGES) project and its working group LandCover6k that in turn received support from the Swiss
National Science Foundation, the Swiss Academy of Sciences, the US National Science Foundation, and the
Chinese Academy of Sciences.
**Acknowledgments**
We are grateful to all palynologists who either contributed origianl pollen counts to this work (Bo Cheng, Yaqin
Hu, Jie Li, Shicheng Tao, YongBo Wang, Ruilin Wen, and Zhuo Zheng) or to the pollen database published by
Cao et al. (2013) from which we used a number of pollen records in this study.

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

©
Harrison, S. P., Gaillard, M. J., Stocker, B. D., Vander Linden, M., Klein Goldewijk, K., Boles, O., Braconnot,
P., Dawson, A., Fluet-Chouinard, E., Kaplan, J. O., Kastner, T., Pausata, F. S. R., Robinson, E., Whitehouse, N.
J., Madella, M., and Morrison, K. D.: Development and testing scenarios for implementing land use and land
cover changes during the Holocene in Earth system model experiments, Geosci. Model Dev., 13, 805-824,
10.5194/gmd-13-805-2020, 2020.
Hellman, S., Gaillard, M.-J., Broström, A., and Sugita, S.: The REVEALS model, a new tool to estimate past
regional plant abundance from pollen data in large lakes: validation in southern Sweden, Journal of Quaternary
Science, 23, 21-42, 10.1002/jqs.1126, 2008a.
Hellman, S. E. V., Gaillard, M.-j., Broström, A., and Sugita, S.: Effects of the sampling design and selection of
parameter values on pollen-based quantitative reconstructions of regional vegetation: a case study in southern

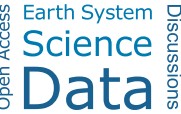

Sweden using the REVEALS model, Vegetation History and Archaeobotany, 17, 445-459, 10.1007/s00334-008-
0149-7, 2008b.
Hou, X.: 1:1 million vegetation map of China, National Tibetan Plateau Data Center [dataset], 2019.
Huntley, B. and Birks, H. J. B.: An atlas of past and present pollen maps for Europe: 0–13,000 years ago. ,
Cambridge: University Press,1983.
Huntley, B. and III., T. W.: HANDBOOK OF VEGETATION SCIENCE, Vegetation history. , Cambridge
University Press, 803 pp.1988.
Kaplan, J. O., Krumhardt, K. M., and Zimmermann, N.: The prehistoric and preindustrial deforestation of
Europe, Quaternary Science Reviews, 28, 3016-3034, http://dx.doi.org/10.1016/j.quascirev.2009.09.028, 2009.
Kaplan, J. O., Krumhardt, K. M., Gaillard, M. J., Sugita, S., Trondman, A. K., Fyfe, R., Marquer, L., Mazier, F.,
and Nielsen, A. B.: Constraining the Deforestation History of Europe: Evaluation of Historical Land Use
Scenarios with Pollen-Based Land Cover Reconstructions, Land, 6, 91, ARTN 91
10.3390/land6040091, 2017.
Klein Goldewijk, K., Beusen, A., Doelman, J., and Stehfest, E.: Anthropogenic land use estimates for the
Holocene – HYDE 3.2, Earth Syst. Sci. Data, 9, 927-953, 10.5194/essd-9-927-2017, 2017.
Klein Goldewijk, K., Beusen, A., van Drecht, G., and de Vos, M.: The HYDE 3.1 spatially explicit database of
human-induced global land-use change over the past 12,000 years, Global Ecology and Biogeography, 20, 73-
86, 10.1111/j.1466-8238.2010.00587.x, 2011.
Li, F., Cao, X., Herzschuh, U., Jia, X., Sugita, S., Tarasov, P., Wagner, M., Xu, Q., Chen, F., Sun, A., and
Gaillard, M.-J.: What do pollen-based quantitative reconstructions of plant cover tell us about past anthropogenic
deforestation in eastern china?, Pages Magazine, 2018a.
Li, F., Gaillard, M.-J., Xu, Q., Bunting, M. J., Li, Y., Li, J., Mu, H., Lu, J., Zhang, P., Zhang, S., Cui, Q., Zhang,
Y., and Shen, W.: A Review of Relative Pollen Productivity Estimates From Temperate China for Pollen-Based
Quantitative Reconstruction of Past Plant Cover, 9, 10.3389/fpls.2018.01214, 2018b.
Li, F., Gaillard, M.-J., Cao, X., Herzschuh, U., Sugita, S., Tarasov, P. E., Wagner, M., Xu, Q., Ni, J., Wang, W.,
Zhao, Y., An, C., Beusen, A. H. W., Chen, F., Feng, Z., Goldewijk, C. G. M. K., Huang, X., Li, Y., Li, Y., Liu,
H., Sun, A., Yao, Y., Zheng, Z., and Jia, X.: Towards quantification of Holocene anthropogenic land-cover
change in temperate China: A review in the light of pollen-based REVEALS reconstructions of regional plant
cover, Earth-Science Reviews, 103119, https://doi.org/10.1016/j.earscirev.2020.103119, 2020.
Li, F. **Gridded pollen-based Holocene regional plant cover in temperate and northern subtropical China**.
National Tibetan Plateau Data Center, DOI: 10.11888/Paleoenv.tpdc.272292. CSTR:
18406.11.Paleoenv.tpdc.272292, 2022.~~.~~
Lu, Z., Miller, P. A., Zhang, Q., Zhang, Q., Wårlind, D., Nieradzik, L., et al. ~~(2018).~~ Dynamic vegetation
simulations of the mid-Holocene Green Sahara. Geophysical Research Letters, 45. https://
doi.org/10.1029/2018GL079195, 2018.
Marquer, L., Gaillard, M.-J., Sugita, S., Trondman, A.-K., Mazier, F., Nielsen, A. B., Fyfe, R. M., Odgaard, B.
V., Alenius, T., Birks, H. J. B., Bjune, A. E., Christiansen, J., Dodson, J., Edwards, K. J., Giesecke, T.,
Herzschuh, U., Kangur, M., Lorenz, S., Poska, A., Schult, M., and Seppä, H.: Holocene changes in vegetation
composition in northern Europe: why quantitative pollen-based vegetation reconstructions matter, Quaternary
Science Reviews, 90, 199-216, http://dx.doi.org/10.1016/j.quascirev.2014.02.013, 2014.
Marquer, L., Gaillard, M.-J., Sugita, S., Poska, A., Trondman, A.-K., Mazier, F., Nielsen, A. B., Fyfe, R. M.,
Jönsson, A. M., Smith, B., Kaplan, J. O., Alenius, T., Birks, H. J. B., Bjune, A. E., Christiansen, J., Dodson, J.,
Edwards, K. J., Giesecke, T., Herzschuh, U., Kangur, M., Koff, T., Latałowa, M., Lechterbeck, J., Olofsson, J.,
and Seppä, H.: Quantifying the effects of land use and climate on Holocene vegetation in Europe, Quaternary
Science Reviews, 171, 20-37, https://doi.org/10.1016/j.quascirev.2017.07.001, 2017.
Mazier, F., Gaillard, M. J., Kuneš, P., Sugita, S., Trondman, A. K., and Broström, A.: Testing the effect of site
selection and parameter setting on REVEALS-model estimates of plant abundance using the Czech Quaternary
Palynological Database, Review of Palaeobotany and Palynology, 187, 38-49,
http://dx.doi.org/10.1016/j.revpalbo.2012.07.017, 2012.
Mazier, F., Broström, A., Bragée, P., Fredh, D., Stenberg, L., Thiere, G., Sugita, S., and Hammarlund, D.: Two
hundred years of land-use change in the South Swedish Uplands: comparison of historical map-based estimates
with a pollen-based reconstruction using the landscape reconstruction algorithm, Vegetation History and
Archaeobotany, 24, 555-570, 10.1007/s00334-015-0516-0, 2015.
Morrison, K., Gaillard, M. J., Madella, M., Whitehouse, N., and Hammer, E.: Land-use classification, Past
Global Change Magazine, 24, 40-40, 10.22498/pages.24.1.40, 2016.





Ni, J., Cao, X., Jeltsch, F., and Herzschuh, U.: Biome distribution over the last 22,000 yr in China,
Palaeogeography, Palaeoclimatology, Palaeoecology, 409, 33-47,
http://dx.doi.org/10.1016/j.palaeo.2014.04.023, 2014.
Ni, J., Yu, G., Harrison, S. P., and Prentice, I. C.: Palaeovegetation in China during the late Quaternary: Biome
reconstructions based on a global scheme of plant functional types, Palaeogeography, Palaeoclimatology,
Palaeoecology, 289, 44-61, 10.1016/j.palaeo.2010.02.008, 2010.
Overpeck, J. T., Webb Iii, T., and Prentice, I. C.: Quantitative interpretation of fossil pollen spectra:
Dissimilarity coefficients and the method of modern analogs, Quaternary Research, 23, 87-108,
http://dx.doi.org/10.1016/0033-5894(85)90074-2, 1985.
Prentice, I. C.: Pollen representation, source area, and basin size: Toward a unified theory of pollen analysis,
Quaternary Research, 23, 76-86, http://dx.doi.org/10.1016/0033-5894(85)90073-0, 1985.
Prentice, I. C. and Webb III, T.: BIOME 6000: reconstructing global mid-Holocene vegetation patterns from
palaeoecological records, 25, 997-1005, https://doi.org/10.1046/j.1365-2699.1998.00235.x, 1998.
Ren, G. and Beug, H.-J.: Mapping Holocene pollen data and vegetation of China, Quaternary Science Reviews,
21, 1395-1422, http://dx.doi.org/10.1016/S0277-3791(01)00119-6, 2002.
Ren, G. and Zhang, L.: A preliminary mapped summary of holocene pollen data for northeast China, Quaternary
Science Reviews, 17, 669-688, http://dx.doi.org/10.1016/S0277-3791(98)00017-1, 1998.
Strandberg, G., Lindström, J., Poska, A., Zhang, Q., Fyfe, R., Githumbi, E., Kjellström, E., Mazier, F., Nielsen,
A., Sugita, S., Trondman, A.-K., Woodbridge, J., and Gaillard, M.-J.: Mid-Holocene European climate revisited:
New high-resolution regional climate model simulations using pollen-based land-cover, Quaternary Science
Reviews, 281, 107431, 10.1016/j.quascirev.2022.107431, 2022.
Strandberg, G., Kjellström, E., Poska, A., Wagner, S., Gaillard, M. J., Trondman, A. K., Mauri, A., Davis, B. A.
S., Kaplan, J. O., Birks, H. J. B., Bjune, A. E., Fyfe, R., Giesecke, T., Kalnina, L., Kangur, M., van der Knaap,
W. O., Kokfelt, U., Kuneš, P., Lata\l owa, M., Marquer, L., Mazier, F., Nielsen, A. B., Smith, B., Seppä, H., and
Sugita, S.: Regional climate model simulations for Europe at 6 and 0.2 k BP: sensitivity to changes in
anthropogenic deforestation, Climate of the Past, 10, 661-680, 10.5194/cp-10-661-2014, 2014.
Stuart, A. and Ord, J. K.: Kendall's Advanced Theory of Statistics. Vol. 1: Distribution Theory. London:
Griffin., 1994.
Sugita, S.: Theory of quantitative reconstruction of vegetation I: pollen from large sites REVEALS regional
vegetation composition, The Holocene, 17, 229-241, 10.1177/0959683607075837, 2007a.
Sugita, S.: Theory of quantitative reconstruction of vegetation II: all you need is LOVE, The Holocene, 17, 243-
257, 10.1177/0959683607075838, 2007b.
Sugita, S., Parshall, T., Calcote, R., and Walker, K.: Testing the Landscape Reconstruction Algorithm for
spatially explicit reconstruction of vegetation in northern Michigan and Wisconsin, Quaternary Research, 74,
289-300, 10.1016/j.yqres.2010.07.008, 2010.
Sun, Y., Xu, Q., Gaillard, M.-J., Zhang, S., Li, D., Li, M., Li, Y., Li, X., and Xiao, J.: Pollen-based
reconstruction of total land-cover change over the Holocene in the temperate steppe region of China: An attempt
to quantify the cover of vegetation and bare ground in the past using a novel approach, CATENA, 214, 106307,
https://doi.org/10.1016/j.catena.2022.106307, 2022.
Tian, F., Cao, X., Dallmeyer, A., Ni, J., Zhao, Y., Wang, Y., and Herzschuh, U.: Quantitative woody cover
reconstructions from eastern continental Asia of the last 22 kyr reveal strong regional peculiarities, Quaternary
Science Reviews, 137, 33-44, http://dx.doi.org/10.1016/j.quascirev.2016.02.001, 2016.
Trondman, A.-K., Gaillard, M.-J., Sugita, S., Björkman, L., Greisman, A., Hultberg, T., Lagerås, P., Lindbladh,
M., and Mazier, F.: Are pollen records from small sites appropriate for REVEALS model-based quantitative
reconstructions of past regional vegetation? An empirical test in southern Sweden, Vegetation History and
Archaeobotany, 25, 131-151, 10.1007/s00334-015-0536-9, 2016.
Trondman, A. K., Gaillard, M. J., Mazier, F., Sugita, S., Fyfe, R., Nielsen, A. B., Twiddle, C., Barratt, P., Birks,
H. J. B., Bjune, A. E., Björkman, L., Broström, A., Caseldine, C., David, R., Dodson, J., Dörfler, W., Fischer, E.,
van Geel, B., Giesecke, T., Hultberg, T., Kalnina, L., Kangur, M., van der Knaap, P., Koff, T., Kuneš, P.,
Lagerås, P., Latałowa, M., Lechterbeck, J., Leroyer, C., Leydet, M., Lindbladh, M., Marquer, L., Mitchell, F. J.
G., Odgaard, B. V., Peglar, S. M., Persson, T., Poska, A., Rösch, M., Seppä, H., Veski, S., and Wick, L.: Pollen-
based quantitative reconstructions of Holocene regional vegetation cover (plant-functional types and land-cover
types) in Europe suitable for climate modelling, Global Change Biology, 21, 676-697, 10.1111/gcb.12737, 2015.
Wyser, K., Kjellström, E., Koenigk, T., Martins, H., and Döscher, R.: Warmer climate projections in EC-Earth3-
Veg: the role of changes in the greenhouse gas concentrations from CMIP5 to CMIP6, Environmental Research
Letters, 15, 054020, 10.1088/1748-9326/ab81c2, 2020.