# Peer review of "Gridded pollen-based Holocene regional plant cover in"

_Earth System Science Data, 2022_

## Author Response (AR1)

**Response to the reviewers' comments**

We wish to express our gratitude to the three reviewers for very useful comments, questions, suggestions, and corrections. They helped us to improve the manuscript significantly.

**RC1**: 'Comment on essd-2022-148', Ralph Fyfe, 05 Sep 2022

The data presented in this contribution are important as they represent a useful step towards quantified vegetation cover across the globe. The results are part of a wider project, and have been presented elsewhere and this paper and accompanying dataset make those data available for the wider community. I agree with the authors that these data will be significant for a broad research community and . The article is well written and describes the data sets, how they were generated, sources of uncertainty, and is suitable for publication as is. There are limitations in the datasets in as much as that groups of 1 degree grid cells carry the same regional vegetation cover estimates, but this is explained and justified by the authors.

I have downloaded and reviewed the datasets. Data are presented in an efficient format for downloading (a series of csv files), with datasets at different level of plant aggregation (plant, pft, land cover type). I have visualised some of the data in QGIS, and it appears to be complete. My only comment on the data download is that there were unusual characters that were interpreted in the download (column 1, grid cell names), which meant I needed to generate additional columns for the easting and northing locations before the data could be projected. Any competent data scientist should be able to manage this process.

These data lie firmly within my research field, and as such I had no difficulty in downloading, understanding and using them.

Answer: Thank you for your positive comment on the utility of the dataset. Re your comment on the data download: we have checked the download but cannot see any unusual characters.

**RC2**: 'Comment on essd-2022-148', Anonymous Referee #2

General comments

This paper describes the use of Holocene pollen records to estimate pollen-based plant-cover reconstruction (27 plant taxa, 6 plant functional types (PFT) and 3 land-cover types (LCT)) for temperate and northern sub-tropical China.

The paper provides information on the Landscape Reconstruction Algorithm approach and its sub-model REVEALS, its current state, data sources and sites, and choice of methods. The results are part of a wider project, the European and northern hemisphere results have already been published. This paper clearly states the limitation of the REVEALS reconstruction, especially the lack of two major taxa (Abies and Picea). The main sources of uncertainty are discussed and future improvements to methods and data are described. Maps provide examples of the LCT reconstructions for major key time windows over the Holocene. The paper is carefully constructed, comprehensive and well explained.

While the paper states that the present the first pollen-based quantitative reconstruction of Holocene plant-cover change, the authors also mention that they use REVEALS estimates previously published in Li et al (2020). Could you further explain the differences between Li et al. (2020) and this paper?

Answer: the objectives of the two papers are different. The main objectives of this paper are to a) make the dataset available in a gridded format useful for modelers, b) describe in details the accuracy and reliability of the REVEALS estimates, as well as their potential applications and limitations. The main objectives of Li et al. (2020) paper were to a) quantify Holocene changes in openland versus woodland cover in the study region and evaluate the role of climate and land use in these changes by comparing the REVEALS reconstructions with earlier interpretations of pollen percentage records, and with palaeoclimate reconstructions and land-use history based on archaeological studies. The REVEALS estimates of individual taxa are the same for the two papers, but grouping of taxa into PFTs and LCTs is different. **The latter is explained in the Method section (L243-250)**. Moreover, visualization is spatial for selected time windows in this paper rather than temporal as in Li et al. (2020). **We also clarified these differences in the Introduction section as follows (L113-115 and L118-124):** "The REVEALS estimates for the 27 plant taxa are exactly the same as in Li et al. (2020), while grouping of taxa into PFTs and LCTs is different. The latter is explained in the Method section below." and "The major differences between Li et al. (2020) and this paper are in the purpose, data visualization, and discussion of the dataset. Li et al. (2020) visualize the results over time for each reconstruction and focus on Holocene changes in openland versus woodland cover and their interpretation in terms of land-use and/or climate-induced changes. The present paper has the major purpose to make the data available to users, in particular climate and vegetation modelers, and explain its potentials and limitations; moreover, it visualizes the results in space and only for a few selected time essentially to provide an illustration of the dataset that says more to the reader than an excel file with numbers."

The main limitation in the datasets is the grouping of sites from adjacent grid cells for the application of the REVEALS model. Although the authors explained and justified their choices, my main concern is not that they grouped sites but rather that they gave results and interpret them at the 1x1° grid cell, see below

Answer: We understand your concern. You seem to understand why we have grouped sites over larger areas than $10^4$ km2 (the assumed special scale of a REVEALS reconstruction). It is done to increase the reliability and decrease the error estimates of the REVEALS reconstructions. We consider this as a limitation because the special scale of the reconstruction is not truly of 1°x1°, but of 1° x1° to maximum 3° x 1°. It means that the resolution of the data is not always of 1°. You worry that we "interpret" the result at a smaller scale. However, we do not "interpret" the results at a 1° grid cell scale; we simply describe changes in plant cover between selected time windows. For each time window and individual taxon (or group of taxa), it is obvious from the maps that the REVEALS estimates are the same for adjacent grid cells. The reconstructions indeed reflect the mean plant cover over more than $10^4$ km$^2$ for large parts of the study region. There are only few grid cells with a REVEALS reconstruction based only on pollen records located within a single grid cell. For several of these grid cells the reconstruction is not reliable because it is based on the pollen record from a single small site or from a large bog. **This is now better emphasized in Fig. 1.** See also our answer to your comments below on the same question.

L231 you have previously said that 57 grids were grouped into 19, and only 18 grids cells remained. It is confusing that results and maps are still expressed into grid cells, while they were not calculated as such.

Answer: we understand that presentation of the data in a gridded format, although the REVEALS reconstructions were not performed at this scale, can lead to confusion. However, we hoped that this was explained clearly enough. **We suggested above a modification of Figure 1 that hopefully clarifies this point.** Moreover, at L 183, the original text was as follows: "The pollen records within 57 of 75 1°x1°grid cells belong to such groups of pollen records (19 in total) from regions larger than a single grid cell. The remaining 18 grid cells include one or two pollen records that could not be grouped with additional pollen records." **We agree that this is not a very clear sentence and propose the following change (from L193 to L202)**: "It implies that, in these cases, the adjacent grid cells (2−8) have the same REVEALS estimates, **i.e. the same mean vegetation cover**. The advantage is that the REVEALS estimates are more reliable and have lower SEs **than REVEALS estimates obtained** for individual 1°×1°grid cells with only one or two pollen records. The pollen records within 57 of 75 1°×1°grid cells belong to such groups of pollen records (19 in total) **covering several grid cells each.** The remaining 18 grid cells include one or two pollen records **only and are emphasized in Figure 1. In these cases, no additional pollen record(s) were available in nearby grid cells, and the REVEALS application was performed with these few pollen records for each grid cell separately**. Several of these reconstructions are based on a single large bog or 1-2 small sites and should

**therefore be considered as less reliable (Fig. 1).**

L 268 Yes no surprise that those grid cells are all reaching low value as sites from all grid cells are used for calculation. This is confusing to give number when in fact the value reflect the group. Is there any other way of showing the results without numbering cells that are indeed carrying the same regional vegetation cover estimates?

Answer: We agree with your concern. We do not exactly "number" the grid cells (in the meaning "giving a number to each grid cell), but it is correct that speaking of "a large number of grid cells" or "three grid cells" showing "such and such" cover or change, may be misleading. **In order to avoid any misunderstanding, we deleted all mentions of grid cells in the descriptions, for instance for OL (L293-316)** "The time window 11.7−11.2 ka BP is characterized by OL cover values >80% in most of northwestern China and the Tibetan Plateau. OL values of 40−60% or 60−80% are found in parts of southwestern China and Inner Mongolia. OL values of 40−60% occur also in the lower reach of the Yangtze River region, and values of 20−40% or 10−20% in northeastern China. The time window 10.2−9.7 ka BP shows an increase in OL cover of 10% in northeastern China, and an increase to 60−80% or > 80% in part of Inner Mongolia and the lower reach of the Yangtze River basin, while a decrease of 20% is seen in part of southwestern China. At 8.2−7.7 ka BP, the OL cover declines in most of the reconstructions, most drastically in parts of the Loess Plateau, central Inner Mongolia and the lower reach of the Yangtze River region, where OL decreased with 20−40%, whilst a decrease of 10−20% is …. etc….".

I do understand why you have presented maps of LCT, I'm wondering if you could have presented a graph that synthesize changes in proportion for each LCT focusing on the groups of sites (instead of grid cells)

Answer: What you are asking for was done in Li et al. (2020), this is not the objective of this paper. See details on objectives in our answer above

*Abies* and *Picea* exclusion

L 296 the exclusion of *Abies* and *Picea* needs further explanation. Their exclusion can already be mentioned and justified in the methods.

Answer: yes, we agree that it should be mentioned in the methods. We have added the following text in methods (L211-218): "Note that, in contrast to Cao et al. (2020), Li et al. (2020) chose to use only RPP estimates obtained from pollen-vegetation datasets collected in temperate China. It implies that two important taxa in northwestern China are missing from the reconstruction, namely *Abies* and *Picea*. Cao et al. (2020) used the RPP estimates of *Abies* and *Picea* from Europe assuming that differences in species between Europe and China would not influence significantly their RPP. As long as this assumption is not tested we decided to keep the principle used in Li et al. (2020) for the dataset we are publishing here.".

Other comments

L 234. Is the results from the ten PFTs used in Li et al. 2020 necessary, as you're saying this classification is not the best one for the purpose of your article

Answer: We are not saying that the PFTs we are presenting are "not the best ones for the purpose of our article". We are saying that the grouping of taxa into PFTs and LCTs in Li et al. (2020) is not appropriate for the use of PFTs or LCTs in climate modelling studies or for comparison with DVM vegetation simulations. Therefore, we made another taxa grouping for this paper, which is clearly explained in our manuscript (L241-248): "In this study we used the PFT classification provided in Table 1 in which Cupressaceae is grouped with *Pinus* as belonging to PFT TeNE (temperate shade-intolerant needle-leaved evergreen trees), Elaeagnaceae, *Castanea*, *Juglans* with broadleaved trees as belonging to PFT TeBS (Temperate shade-tolerant broadleaved summer green trees), and Cyperaceae, Poaceae, Rosaceae, and Rubiaceae with all herbs as belonging to PFT C3H/OL (C3 Herbs/openland). We propose that this classification is more appropriate for use in climate modelling contexts than that used in Li et al. (2020) in which the major aim of the study was to interpret the pollen-based plant-cover reconstruction in terms of regional vegetation cover.". **However, in order to avoid misunderstanding, we changed "In this study,…." by "In this paper, …..", and "in terms of regional vegetation cover." By "in terms of open land versus woodland cover."**

L78 LC and LU instead of LU and LC

Answer: We do not understand this comment. The abbreviation "LULC" is commonly used for "anthropogenic land-cover change due to land-use change". However, see our answer to the last reviewer, below

L132 rephrase to the basin type (lake or bog) and its size (area and calculated radius)

Answer: OK, but we had to keep the same order as in the metadata table, i.e. "…, the site size (area and calculated radius) and type (lake or bog), …. ".

L196. You mention three computer programs, only one is specified.

Answer: This is indeed confusing. The calculation steps implemented by the three computer programs are now implemented by a single program. Therefore, we deleted the sentence mentioning three computer programs.

L189 Use either SD or SE and harmonize through the text. I'm not sure that there are differences between the two terms in your results. ERV results are expressed in SE, the delta-method as well.

Answer: Standard deviation and standard error are two different things in mathematics/statistics. It is correct to use standard deviation for the relative pollen productivities (e.g. Li et al., 2017), and standard error for the uncertainties on the REVEALS estimates (Sugita, 2007a).

L203 space between lake(s)) and are

Answer: Thank you! Corrected.

L204 space between obtained and using

Answer: Thank you! Corrected.

L236 10.7-11.2 ka BP instead of 9.7 – 10.2 ka BP

Answer: We do not understand this comment, and cannot find any error in our time windows as mentioned in the text.

L261 Harmonize names of time windows between text and figures

Answer: Thank you! Corrected.

L412. I guess there are ways of testing the impact of RPP's SE on the REVEALS reconstruction, by testing the sensitivity of the REVEALS model using simulation. But this was out of scope of this paper.

Answer: "The standard deviations (SD) of the RPPs are taken into account in the calculation of the REVEALS standard errors (SEs), as well as the number of pollen grains counted in the sample (Sugita, 2007a). The uncertainties in the averaged REVEALS estimates of plant taxa for a grid cell are calculated using the delta method (Stuart and Ord, 1994) and expressed as the SEs derived from the sum of the within- and between-site variations in the REVEALS results in the grid cell (see e.g. Githumbi et al. (2022). We have explained this better in the Method section (L225-228).

L 433 To be provocative I would say "This is a serious caveat if the taxa for which no RPP values are available represent a significant part of the **vegetation** and are present in the pollen assemblage" We want to be able to reconstruct key dominant species in the **vegetation**…Apart *Abies* and *Picea*, are they other dominant taxa in the pollen assemblages with no RPP values?

Answer: We have corrected the first sentence as follows, Line452-454: "This is a serious caveat if the pollen taxa for which no RPP values are available represent a significant part of the pollen assemblages.". You are not more provocative than us, as we write below in the manuscript "In this first dataset of REVEALS land-cover estimates, our decision to use exclusively Chinese RPPs and, therefore, not reconstruct the cover of *Abies* and *Picea* **is a major issue**. This may bias the results in

overestimating the cover of C3H/OL in particular, but also of BT. The latter needs to be kept in mind in the interpretation and use of the dataset **for regions where *Abies* and *Picea* were common during part of, or the entire Holocene**, etc….".

There are no other major pollen taxa (that can be dominant in the pollen record) having no available RPP values. However, for some pollen records, the percentage of pollen taxa for which we do not have RPP values can be high. "The 27 taxa included in this REVEALS reconstruction account for >50% of the total pollen from all pollen taxa in all records, and for > 80% of the total pollen from all pollen taxa in most records." We added this information in Methods (L215-217).

L 443 remove "estimate the **the** past"

Answer: Thank you! Corrected.

L457 remove "be cautious and **The** is a limitation of the"
Answer: Thank you! We have rewritten the sentence as follows (L478-L481): "The interpretation of past changes in openland cover needs to take into account the issues described above. This is a limitation of the gridded REVEALS land-cover dataset if used for validation of ALCC scenarios and studies of human-induced land-cover change as a climate forcing."

L. 469 correct "REVEALS estimates **rby** increasing"

Answer: Thank you. Corrected.

L468, your strategy of having two pollen counts per time window prevent the problem to only have one pollen count documenting the whole period. This pollen count could be linked with a unique event

Answer: Yes indeed. However, the main reason for using time windows is the need of increasing the pollen count used in the REVEALS application. A single pollen count is too small and will result in large standard errors on the REVEALS estimates. We have explained this earlier in many papers (e.g. Trondman et al. (2015), Li et al. (2020) etc.).

L500 insert a space between "cover_(e.g."

Answer: Thank you. Corrected.

L 516 the information on "x years (x years is the number of years between 1950 CE and the year of coring)" should be already mentioned in the methods

Answer: We have edited the description in methodology from line 175 "the length of the three most recent time windows were fixed to 350 (0.7−0.35 ka BP), 250 (0.35−0.1

ka BP), and 100 + x years (0.1 ka BP to present (1950 CE + x years, where x years is the number of years between 1950 CE and the year of coring).

Figures

- Fig 1: is the graph in the right lower panel useful?

Answer: All maps of China for publication purpose must be approved by Chinese law which requires that the Spratly islands are included in Chinese territory (http://bzdt.ch.mnr.gov.cn/ ). As a Chinese citizen, the first author of the paper has to follow the law.

- Fig 2-4 could you improve the maps by adding key information that you're referring in the text (i.e. Yangtze River region, vegetation zone…)

Answer: the information on vegetation zones and location of the Yangtze River is available in figure 1. Adding them again in figures 2-4 will make them too messy.

- Fig 4, ensure error circle of grid cell 30°N-90°E is in the reconstructed cell for time window 5.7 – 6.2 ka BP

Answer: Thank you! We have removed the circle from the figure, it was a mistake.

**RC3**: 'Comment on essd-2022-148', Anonymous Referee #3, 21 Sep 2022  reply

**Review of ESSD-2022-148 by Li et al.**

This paper describes a new REVEALS data set of vegetation in China during the Holocene. As this data set is the first of its kind it's a welcome contribution for everyone interested in the vegetation and climate of the Holocene. I think that the paper describes the data set and methods well.

I have only one major concern, and that is about how the word "gridded" is used here. I understand that gridded information here means that several data point together represent a larger area, a grid cell, instead of being just point data. On the other hand, when I hear about a gridded data set I expect it to cover a larger region. If the grid contains more empty than filled grid cells it's not that different from point data. I understand the problems of creating such a data set for the Holocene in China. I think that it's perfectly fine to publish a data set like this, but I think that the authors should mention this, especially since you write that the data could be used in climate model simulations. The use for such a data set is limited in a climate model. The data can be used to evaluate results from climate or vegetation models, but it can't be used in a climate model simulation.

Answer: **We agree with the reviewer, and have clarified this as follows:**

**In the Introduction (L90-96)** "Given that the gridded REVEALS reconstructions are not continuous over space, i.e. only a part of the grid cells have pollen-based REVEALS estimates of plant cover, such a dataset is comparable to a collection of point data in space. It implies that the REVEALS data need to be interpolated over space to produce a true gridded dataset with values of plant cover in all grid cells. Such interpolations were performed using the European gridded REVEALS reconstructions (e.g. Pirzamanbein et al., 2012; Githumbi et al., 2022; Strandberg et al., 2022) and used for the first time in climate modelling by Strandberg et al. (2022).".

**In the Discussion's last section (L513-518)** "They can be used for various purposes, such as the evaluation of scenarios of past deforestation (HYDE and KK) (Kaplan et al., 2017) or comparison with simulations of past vegetation cover using dynamic vegetation models (Marquer et al., 2014, 2017). For use in climate modeling experiments looking into e.g. past human-induced land cover (or land use) as a climate forcing, the REVEALS plant-cover data need to be interpolated over all grid cells of the simulation geographical domain using for instance spatial statistics (e.g. Strandberg et al., 2022; see also the Introduction section)."

*Comments*

L140: Maybe it would be good to explain that the grey colours are fill colours in the boxes and not the grey lines mentioned earlier.

Answer: we have edited the sentence as follows (L152-154): "Grid cell reliability in terms of REVEALS estimates of plant cover is indicated by fill colours, light grey for high reliability and dark grey for low reliability, as defined on the basis of the number and type of pollen sites (see text for detailed explanations)."

L155: "from 1 large and small sites." What does that mean?

Answer: we deleted "l", this was a mistake!

L165: "350, 250 and 100 years" How does this fit with the requirement of 2 counts/500 years (L127)? Does it mean that the temporal resolution is higher in the last years, or that one point represents several time periods?

Answer: Yes, the temporal resolution is generally higher over the last 1000 years, which is explained in the methods (L177-180).

L180: "several adjacent grid cells (2-8)" Maybe I just don't get this, but I cant see more than 5-6 adjacent grid cells in figs 2-5.

Answer: This was indeed expressed in a confusing way. We changes the sentence as followed (L193-195): "It implies that, in these cases, the grid cells covered by a group of pollen sites (varies between 2 and 8 grid cells, Fig. 1) have the same REVEALS estimates, i.e. the same mean vegetation cover (Figures 2-4)."

L189: "atmospheric conditions" What's atmospheric conditions more than wind speed, and why is wind speed not considered an atmospheric condition? Please explain.

Answer: In the context of pollen-vegetation modeling, atmospheric conditions refer to the movements of air masses in the atmosphere that can influence pollen transport in the air, wind speed included. In the models pollen-vegetation modelers are using, atmospheric conditions include the parameters Cz, Cy, n and u. Cz and Cy are vertical and horizontal diffusion coefficients, n is a dimensionless turbulence parameter, and u is wind speed (expressed in m/sec). The values of Cy, Cz, and n depend on atmospheric stability. If neutral conditions are assumed, the values of the three parameters are 0.12, 0.21, and 0.25, respectively. We do not explain this in detail in the paper as it has been described in several papers earlier. **We refer instead to one of the major paper on the subject, Jackson and Lyford (1999), as follows (L206-207)**: "and atmospheric conditions (expressed by four parameters, i.e. vertical and horizontal diffusion coefficients, a dimensionless turbulence parameter, and wind speed (see Jackson and Lyford, 1999 for details).".

L245-247: I don't get this at all. What does it mean that the "map changes are expressed in comparison to the former"? One interpretation of this is that 80 % means 80 % of the cover in the previous time slice. If the first time slice shows 50 % and the second 80 % that would in fact mean 80 % of 50 % = 40 %. I don't think this is the case, because it would make it extremely complicated to calculated what the vegetation fractions are in 100 BP. Also, since the legends in figs 2-4 show 0-100 % I think that the absolute fractions are what is show. Otherwise the scale would include negative values or values over 100. Please rephrase or explain.

Answer: Yes, you are perfectly correct, what we show in the map are absolute fractions, and we describe these maps in the text using absolute fractions. We have been thinking of the confusion the use of absolute fractions in a text might lead to. Thank you for the solution you suggest to us, this is definitely the best way to explain this. We changed the text as follows (L274-279 ): "For each land-cover type the maps are described from the oldest (11.7–11.2 ka BP) to the youngest (0.1 ka–present) map, and the map for each time window is described in comparison to the map for the earlier time window (e.g. for the 9.7–10.2 ka BP map, changes are expressed in comparison to the 11.7–11.2 ka BP map). Land-cover changes (decrease or increase) are expressed in absolute fractions, e.g. an increase of 20% at 9.7-10.2 ka BP compared to a cover of 50% of the grid cell at 11.7-11.2 ka BP implies that the cover at 9.7-10.2 BP is 70% of the grid cell.".

L489-496: As said above, in its present form this data would be complicated to use in a climate model. That would require a denser grid. Maybe you could say something about that, or say that it's a long term goal to make it work in climate models.

Answer: YES, we agree. See answer to your comment above.

*Typos*

L63: "Iii" I think it should be "III".

Answer: done, thank you

L69: LULC usually means more then land-use change as it would be interpreted here. Please spell out land-use and land-cover changes, if that's what you mean.

Answer: LULC should be understood as the abbreviation of what precedes it in the sentence, i.e. "anthropogenic land-cover (LC) transformation due to past land-use (LU) change" (or simply land-cover change due to land-use change, "LU change-induced LC change", LULC, commonly used in the literature). However we have corrected LULC into "LULC change" (L69, L72, and L89).

L204: "obtainedusing" -> "obtained using"

Answer: corrected, thank you

L457: "and The is a" Something is wrong with this sentence.

Answer: corrected, thank you

L469: "rby" -> "by"

Answer: corrected, thank you

L491: "Iii" -> "III"

Answer: corrected, thank you

L496: "coverl" -> "cover"

Answer: corrected, thank you

---

## Author Response (AR2)

Li et al.: Gridded pollen-based Holocene regional plant cover in China

**Response to Topic Editor Comments**

Due to adverse formatting, reference very hard to read and check. For example, Githumbi et al. 2022, often cited in text, not included in reference list. Possible that these errors occur too many times. Please completely reformat reference list in clean useable version then carefully check all references.

Response: We apologize for the inconvenient format of our reference list. It is now reformatted and a line space between references was added. Githumbi et al., 2022 was indeed missing, we have added it and all citations are now crosschecked. Several other references were missing in the references list, and now added. Moreover, there are two different papers Githumbi et al. 2022 that are both referred to in our text. They are now 2022a and 2022b.

For copyright reasons, journal will need source information for every satellite image used a background (e.g. in Figure 1). Relevant to reviewer comment, Copernicus will accompany this publication with a disclaimer about territorial (Spratly Islands) claims.

Response: We have added the source information in the figure caption. We agree that Copernicus accompanies this publication with a disclaimer about territorial claims.

I predict Copernicus image experts will object to panel B of Figure 1. Colors (light grey, dark grey) and lines (thick vs thin) not obvious to this readers. Authors should build improved version.

Response: see response below under line 199.

What does the diagonal purple dashed line indicate?

Response: We have added the information in the figure caption.
"The diagonal pink dashed line indicates the modern Asian summer monsoon limit according to Chen et al. (2010)"

Line 163: "northern America" not a proper geographic reference. Northern USA? Canada?

Response: we have corrected into "Northern USA".

Line 199: "emphasized in Figure 1" Unfortunately, not clear for this reader.

Response: we have changed the light grey into white and the dark grey into black. The thick black lines are now all similar circles. It should now be easier to distinguish the

lines thickness and colors (see revised Figure 1 B and its revised caption). Moreover we have rewritten lines191-204 to clarify all the section, as follows: "Due to the low spatial density of the 94 selected pollen records in this study, the pollen records were grouped for the application of the REVEALS model within coherent regions with comparable biogeographical characteristics and similar vegetation histories (see Li et al. (2020) for details). It implies that, in these cases, the grid cells covered by a group of pollen sites (varies between 2 and 8 grid cells, Fig. 1) have the same REVEALS estimates, i.e. the same mean vegetation cover (Figures 2-4). This is a deviation from the standard protocol used in Europe for which pollen records were never grouped within more than a single 1˚×1˚ grid cell. The reason for grouping pollen records over more than one grid cell (18 groups of grid cells, 57 of 75 grid cells in total) was to increase the reliability of the REVEALS estimates in areas with sparse distribution of pollen records.  The remaining 18 grid cells are isolated, i.e. no additional pollen record(s) were available in nearby grid cells, and the REVEALS application wasperformed for each grid cell separately. Eight of these grid cells include one or two large lakes and provide reliable REVEALS reconstructions of plant cover. The other 10 grid cells (emphasized by a thick black circle in Figure 1B) include  one or two small site and represent therefore  reconstructions that need to be considered with caution, of which five are based on one small site only and labelled as less reliable (black grid cells in Figure 1B)."

Line 202: How much "less reliable"? Are these uncertain data flagged somehow, somewhere?

Response: See revision above

Figures 2, 3 and 4: Circles, filled or unfilled, or other textures within small squares prove very obscure to these old eyes. Again, I suspect Copernicus image experts will object. Authors should consider alternate formats to convey same information?

Response:  We have done the best we could with this and cannot think of a better way to visualize the standard errors of the REVEALS estimates of plant cover. Githumbi et al. (2022) also used this system to show the errors (and had blue circles as well) in a similar paper on gridded REVEALS reconstructions in Europe published in ESSD; this was obviously accepted by Copernicus. Moreover, we are of the opinion that such figures can be read on the computer where it is possible to zoom on the grid cells of interest and clearly see the circles, whether they fill the cell or are smaller.

NOTE:
**Besides the revisions above**, we have made a few other revisions (mainly language editing and clarifications) that are also emphasized in the revised manuscript.